# Ecoenergetic Comparison of HVAC Systems in Data Centers

**Alexandre F. Santos [1,2], Pedro D. Gaspar [1,3,\*]**  **and Heraldo J. L. de Souza [2]**

1 Department of Electromechanical Engineering, University of Beira Interior, 6201-001 Covilhã, Portugal; d1682@ubi.pt
2 FAPRO—Professional College, 80230-040 Curitiba, Brazil; heraldosouza1@gmail.com
3 C-MAST—Centre for Mechanical and Aerospace Science and Technologies, 6201-001 Covilhã, Portugal
\* Correspondence: dinis@ubi.pt; Tel.: +351-275-329-759

**Abstract:** The topic of sustainability is of high importance today. Global efforts such as the Montreal Protocol (1987) and the Kigali Amendment (2016) are examples of joint work by countries to reduce environmental impacts and improve the level of the ozone layer, the choice of refrigerants and air conditioning systems, which is essential for this purpose. But what indicators are to be used to measure something so necessary? In this article, the types of air conditioning and GWP (Global Warming Potential) levels of equipment in the project phase were discussed, the issue of TEWI (Total Equivalent Warming Impact) that measures the direct and indirect environmental impacts of refrigeration equipment and air conditioning and a new methodology for the indicator was developed, the TEWI DC (DC is the direct application for Data Center), and using the formulas of this new adapted indicator it was demonstrated that the TEWI DC for Chicago (USA) was 2,784,102,640 kg $CO_2$/10 years and Curitiba (Brazil) is 1,252,409,640 kg $CO_2$/10 years. This difference in value corresponds to 222.30% higher annual emissions in Chicago than in Curitiba, showing that it is much more advantageous to install a Data Center in Curitiba than in Chicago in terms of environmental impact. The TEWI indicator provides a more holistic view, helping to combine energy and emissions into the same indicator.

**Keywords:** global warming; TEWI; Data Center; climate; refrigerants; EUED

## 1. Introduction

Brazil entered the Montreal Protocol in 1990 with an emphasis on eliminating chlorofluorocarbon (CFC) refrigerants. The National Seminar "Government and Society on the way to the elimination of HCFCs" held in 2009 marked the beginning of the elaboration of the Brazilian hydrochlorofluorocarbons (HCFC) Elimination Program (PBH). This document defines the guidelines and actions to be carried out in Brazil related to meeting the goals of eliminating the consumption of HCFCs.

In the context of air conditioning and refrigeration, the program states that by the year 2040 the consumption of HCFCs will be eliminated. The Kigali Amendment set the deadlines for the elimination of HFCs (hydrofluorocarbons). Thus, it is necessary to improve the technologies with less of an environmental impact. The market trend is the migration to natural refrigerants, such as ammonia (R-717), carbon dioxide (R-744), and hydrocarbons. Despite having low global warming potential (GWP) rates, these refrigerants can be toxic, flammable, or work only with high pressures, requiring more careful inspections and the adoption of adequate safety measures and standards [1].

The Leadership in Energy and Environmental Design (LEED) certification proposed by the US Green Building Council (USGBC), contains a topic in the energy and atmosphere scope, where it is possible to add one point to the mark considering the refrigerant fluids. It is called Enhanced refrigerant management, where the cooling fluids of a building are subjected to a method of calculation of the refrigerant impact. In this method, refrigerant fluids that are used in heating, ventilation and air-conditioning, and refrigeration (HVAC&R)

equipment must minimize or eliminate the emission of compounds that contribute to ozone depletion and climate change, given by the Global Warming Potential (GWP) [2].

This method only applies to GWP and ODP (Ozone Depletion Potential), but does not focus, for example, on the association of energy with global warming. In addition, the GWP caused by refrigerant fluids in a green building punctuates only one point, with no line basis for the so-called "green buildings". Scofield and Cornell [3] stated that in the certification process they "do not account for energy losses outside the building boundary associated with producing fuels and transporting them to the building. These off-site energy losses may not be of interest to the building owner, but they are very important when looking at energy policy, total resource consumption, energy costs, and the environmental impact of a building".

This article has two correlated objectives: (1) to develop GWP comparisons for different HVAC solutions, for the same thermal load to raise awareness of the importance of implementing global warming indices when deciding to choose which type of HVAC system; (2) to create an eco-energy comparison methodology to apply the sum of electric energy and global warming in Data Centers (DCs) using the superposition of two methods, TEWI (Total Equivalent Warming Impact) and EUED (Energy Usage Effectiveness Design). With the superposition from these comparison methods it will be possible to compare the eco-energetic behavior of an identical HVAC system in different cities and energy sources, allowing the proposal of a new comparison indicator useful for DC, the TEWI DC.

In this research, comparisons and indicators were created to facilitate the replication of the technological solutions adopted. The $kgCO_2$/Refrigeration ton indicator shows how each system, with its respective refrigerant fluid, acts on global warming according to the thermal load. The TEWI indicator was used to compare results between the national (Brazil) and other countries markets, thus showing the impact of fluids in each market, considering the countries' energy mix.

The TEWI indicator has the advantage of measuring global warming impacts, directly and indirectly, so it is possible to perform sensitivity analysis for each world region and each kind of application [4].

The Standard 90.1-2019 [5], as energy standard for buildings except low high residential buildings, is the main document for comparing the energy efficiency with a baseline for buildings, with indicators even in partial loads such as IPLV (Integrated Part Load Value). This study contributes in this sense, extending the indicators to an eco-energy approach [5,6].

A case study, including the topics highlighted in the research, was developed for a DC project. A DC is a facility that brings together servers, processing equipment, physical or virtual, for the processing, storage, management and distribution of data for a given business. The air conditioning systems must be able to operate the 8760 uninterrupted annual hours for this type of facility. Due to their growing global importance, DCs are becoming infrastructures where it is important to test and apply new methods to reduce impacts on the environment and improve their energy efficiency [7]. The place where a DC is installed can also influence these indicators by its thermal characteristics and also by the energy matrix. Therefore, two cities with similar thermal characteristics will be chosen for a comparison of ecoenergy indicators and to verify the direct and indirect environmental impacts of the system, the choice of the type of the DC building is basically related to two reasons. One concerns the need of growing the data storage and another is due to the constant thermal load.

## 2. Methodology

### 2.1. Global Warming Potential (GWP)

Environmental concerns have become a driving force in efforts to optimize green projects through increased energy efficiency, research into new refrigerants, and efficient use of old systems. GWP is the metric used by the IPCC (Intergovernmental Panel on

Climate Change) to compare the potential climatic impact of emissions of different LLGHGs (long-lived greenhouse gases). Each refrigerant fluid has an established GWP value [8].

The impact is estimated over a period. The 100-year time horizon is the most adopted and it is usually assumed when no information in the time horizon is given. GWP is an easy-to-use metric, because the lower the GWP, the less a substance contributes to global warming [9].

The refrigerants most used in air conditioning are R-22, R 134 A, R 410 A, which have a GWP of 1810, 1430 and 2088 respectively [8].

### 2.2. Total Equivalent Warming Impact (TEWI)

TEWI is a metric of the global warming impact of the equipment based on the total emissions related to GWP during the operation of the equipment and the removal of operational fluids at the end of life. TEWI considers both direct and indirect emissions generated through the energy consumed in the process of the equipment. TEWI is measured in kg of carbon dioxide equivalent [10]. TEWI is calculated with the sum of two parts:

1. Direct Emission—Refrigerant released during the life of the equipment, including losses not recovered on the final disposal.
2. Indirect Emission—The impact of $CO_2$ emissions from fossil fuels used to generate the electric energy that is used in the operation of the equipment throughout its life.

The method of calculating TEWI is provided in Equations (1) and (2):

$$\text{TEWI} = \text{GWP}(\text{direct, refrigerant leaks including EOL}) + \text{GWP}(\text{indirect, operation}) \tag{1}$$

$$\text{TEWI} = (\text{GWP} \cdot L_{annual} \cdot n) + \text{GWP} \cdot m \cdot (1 -_{recovery}) + (E_{annual} \cdot \beta \cdot n) \tag{2}$$

where:
EOL = End of Life.
GWP = Global Warming Potential of refrigerant, relative to $CO_2$ (GWP $CO_2$ = 1)
$L_{annual}$ = Leakage rate p.a. (kg).
$n$ = System operating life (yrs).
$m$ = Refrigerant charge (kg).
$\alpha_{recovery}$ = Recovery/recycling factor from 0 to 1.
$E_{annual}$ = Energy consumption per year (kWh p.a.).
$\beta$ = Indirect emission factor (kg $CO_2$/kWh) [10].

GWP is an important input data for measuring TEWI, but TEWI is a more complete indicator, as it also addresses the issue of gas leaks and indirect emissions from the energy matrix.

The initial cost of an air conditioning system is important for the investor, but the cost of operation over time becomes a more complete indicator for decision making. The TEWI value reaches in the same formula the total energy and emissions in the life cycle.

The results point out that refrigerant fluids have, proportionally, a greater impact in Brazil than in other important world markets.

The indirect emission factor, $\beta$, given in kg $CO_2$/kWh, varies according to the energy matrix. For example, in Brazil the matrix of the energy system is shared in the same transmission for the whole country. According to National Energy Balance Brazil (BEN) [11], Brazil emits 0.088 kg $CO_2$/kWh. According to the Energy Information Administration (EIA) [12], the United States of America emits 0.417 kg $CO_2$/kWh.

### 2.3. Data

The numerical data for the GWP was taken from the IPCC AR-04 [8], while the refrigerant charge data was used as recommended by the manufacturers, specifically for systems with the largest refrigerant piping. All VRF systems were compared using LATS Heating, Ventilation, and Air Conditioning (HVAC) software. The result of a decreasing value of $CO_2$ emissions by the system is shown in Table 1 and shown in Figure 1 [13–15].

*Comparison of GWP Values and Absolute CO$_2$ Emissions for Different Solutions for the Purpose of Cooling the Same Thermal Load. Issuance of Installed Equipment, without Operation (Inactive)*

The thermal load was performed considering a 5-stage generic DC with 400 TR of thermal load. The Ton Refrigeration (TR) unit is often used as a general term to indicate the capacity or size of a refrigeration plant. It is defined as the rate of heat transfer to freeze (or melt) 1 ton (907 kg) of pure ice at 0 °C in 24 h. ASHRAE defines 1 TR as equivalent to a cooling capacity of 3516.85 W or 3023.95 kcal/h. The systems chosen for comparison are available in the Brazilian market. The analyzed systems were:

1.   Variable Refrigerant Flow (VRF) of air installed on the roof and serving all floors;
2.   Air VRF—placed on each floor to reduce the amount of refrigerant piping;
3.   VRF water;
4.   Air chiller;
5.   Water chiller;
6.   Water chiller with LOW-GWP technology;
7.   Window type equipment;
8.   Conventional split

**Table 1.** Comparison of the amount of CO$_2$ emissions for each type of system.

| System | Brand Ref. | Model Ref. | Fluid | GWP | Fluid (kg) | CO$_2$ (kg) | CO$_2$ (kg/Ton) * |
|---|---|---|---|---|---|---|---|
| VRF Air Cooled Coverage | LG | ARUM500LTE5 | R-410A | 2088 | 510 | **1,064,880** | 2662.20 |
| VRF Air Cooled Floor | LG | ARUM500LTE5 | R-410A | 2088 | 499 | **1,041,912** | 260.78 |
| Split | Daikin | STK12P5VL | R-410A | 2088 | 360 | **751,680** | 188.00 |
| Air Cooled chiller | Carrier | 30XAB400 | R-134a | 1430 | 315 | **450,450** | 1126.13 |
| Window air conditioning | Gree | GJC12BL | R-22 | 1810 | 240 | **434,400** | 1086.00 |
| VRF Water cooled | LG | ARWN800LAS4 | R-410A | 2088 | 201 | **419,688** | 1049.22 |
| Water cooled chiller | Carrier | 30XWB400 | R-134a | 1430 | 245 | **350,350** | 876.00 |
| Water cooled chiller Low GWP | Johnson Controls | YZ-MA041AN0 | R-1233zd | 1 | 395 | **395** | 0.99 |

\* CO$_2$ (kg/Ton) is the amount of emission per ton of refrigeration (equipment off, but with refrigerant charge).

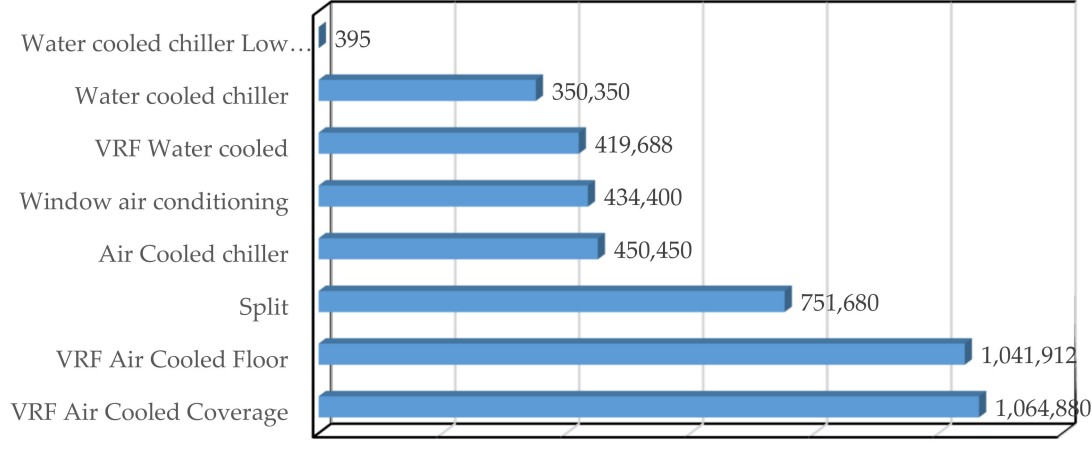

CO$_2$ Emissions [kg]

**Figure 1.** CO$_2$ emissions by type of system.

It is possible to observe that the "Water chiller low GWP" emits less than 0.1% of the $CO_2$ of the most polluting system, the VRF air installed on the roof and that serves all floors. This is due to the refrigerant used, the R-1233zd, a recent family of HFOs (hydrofluoroolefins), which has ultra-low GWP (GWP = 1), high efficiency, safety rating A1 (non-toxic and non-flammable), and has great availability. It is a low pressure fluid that is similar in performance to the R-11 but has the advantage of slightly higher pressures. According to the IEA [14], the VRF and split systems represent 82% of the air conditioning equipment in the world, while the units with the lowest refrigerant charge, which are the chillers and packaged units, are only 18%. Thus, even though VRFs are energy efficient, there should be a rethink about the types of air conditioning systems to be used when it comes to sustainability from the refrigerant point of view [14].

Despite the growth in sales of the VRF system worldwide, often due to the ecological appeal of energy savings, it is evident that it is the system with the highest refrigerant charge, which in turn has an enormous level of kg $CO_2$ emissions. Something important to note is that from a refrigerant charge point of the view, window air conditioners use less charge than VRF systems, and the welds of the pipes are made in the factory of the window air conditioner while they are manually VRF and split.

## 3. Case Study

This study contrasts the results of the comparison metrics for a high-density data center placed in cities with similar characteristics. These are:

- Curitiba—Brazil;
- Chicago—United States of America (USA).

For the eco-energy comparison in DC, two cities with similar thermal characteristics were used in opposite hemispheres, one of them Curitiba, where 7894 h per year (according to ASHRAE Weather Data Viewer) are below 25° C, and the other, Chicago, where for the same characteristic and data source, the number of hours below 25°C is 7628 h, that is, a difference of only 3% (both dry bulb temperatures).

The dry bulb temperature of Chicago is 6.37% higher than Curitiba. For psychometric effects, the altitudes of Curitiba and Chicago are 908 and 182 m, respectively [16,17]. Table 2 shows the data of an average of the hottest hours of the year, frequency year [%], dry bulb (DB) temperature (°C), wet bulb (WB) temperature (°C), and altitude (m) for the cities of Curitiba and Chicago.

**Table 2.** Data for the cities of Curitiba and Chicago (adapted from [16,17]).

| Cities | Frequency year (%) | DB Temperature (°C) | WB Temperature (°C) | Relative Humidity (%) | Enthalpy (kJ/kg) | Altitude (m) |
|--------|--------------------|--------------------|--------------------|--------------------|--------------------|-------------|
| Curitiba | 0.4 | 30.9 | 23.2 | 53.4 | 74.14 | |
| | 1.0 | 29.8 | 22.6 | 55 | 71.67 | 908 |
| | 2.0 | 28.7 | 22.0 | 57 | 69.2 | |
| Chicago | 0.4 | 33.0 | 25.4 | 54 | 78.8 | |
| | 1.0 | 31.5 | 24.4 | 56 | 74.2 | 182 |
| | 2.0 | 30.0 | 23.5 | 58 | 70.9 | |

The case study for comparing the metrics was applied to a high-density DC of $28.8 \times 28.75$ m, with equipment for heat dissipation (695.52 kW of sensible heat, that is, 0.84 kW/m$^2$) added to the internal thermal load of 96 kW, composed by driving and lighting loads (16 kW) and people and others (80 kW of losses in No break and others). For this thermal load condition, 16 pieces of equipment were selected (8 effective and 8 reserves) of 30 TR "dual fluid" each. The dual fluid equipment, LIEBERT L99 [18], had the technical specifications shown in Table 3.

**Table 3.** Technical specifications of the Dual Fluid equipment (adapted form [18]).

| Model | Unit | Value |
|---|---|---|
| Total cooling power | kW | 104.5 |
| Sensible cooling power | kW | 88.7 |
| Sensible Heat Rate | | 0.85 |
| EER | kW/kW | 3.40 |
| Number of compressors | n | 2 |
| Air delivery | $m^3/h$ | 21,100 |
| Maximum available static pressure | Pa | 90 |
| Sound pressure level | dB(A) | 66.9 |
| Width | mm | 2550 |
| Depth | mm | 890 |

A nominal Energy Efficiency Rate (EER) of 3.4 kW/kW is obtained for the approximate cooling power of 30 TR (104.6 kW for each of the sixteen equipment). This EER value considers a total nominal equipment consumption of 30.86 kW, for the air inlet temperature condition of 35 °C in the condenser. The AHRI 1361 standard (2017) [19] defines the energy efficiency data for DC air conditioning equipment. The nominal consumption of the air conditioning is 246.88 kW to supply the thermal load. The most used metric to describe how efficiently a DC uses energy is PUE (Power Usage Effectiveness). This metric is the ratio of the total amount of energy applied by a DC facility to the energy delivered to computing equipment. For this case study, the PUE, given by Equation (3), comes:

$$PUE = \frac{\text{Total Energy (IT equipment + air conditioning + Lighting + UPS losses and inverters and others)}}{\text{Energy of IT equipment}} \quad (3)$$

$$PUE = \frac{(694.60 + 246.88 + 16.00 + 80.00)}{(694.60)} = 1.50 \, [\text{kW/kW}]$$

The DC facility is the most energy efficient as it is lower than the PUE value. The value of PUE = 1.50 kW/kW for this case study is excellent. A typical DC has a PUE of around 2.1 kW/kW [2].

The CoolPack software [20] was used to determine the compressor's isentropic coefficient. For this purpose, the power of the evaporator and condenser fans were neglected. Thus, the EER value was updated from the catalog condition (=3.4 kW/kW including the condenser and evaporator fans), discarding fan power (7.36 kW): EER = 104.6 kW/(30.86 kW − 7.36 kW) = 4.5 kW/kW (compressor specific).

## 4. Comparison of Energy Efficiency Metrics

The power usage effectiveness Constant External Air (PUE COA) is the energy efficiency metric related to the use of the COA temperature on average 0.4% of the current highest temperatures of NBR 16401, 2017 [15]. In the design phase, the PUE is a metric between electrical powers and is an indicator widely used to compare one DC with another.

The Energy Usage Effectiveness Design (EUED) index shown in [21,22] is determined in this study. It uses the weather data for 8760 h per year for a given city, using the wet and dry bulb temperature data for those 8760 h. The method considers psychrometric conditions for the use of systems such as free cooling, evaporative system, and variable COPs (Coefficient of Performance). Applying the method with the data from the ASHRAE Weather Data Viewer, it is possible to know how many hours per year each air conditioning solution is applied in a specific region to install a DC. EUED allows a comparison in the design phase that would only be possible in the field with the Datacenter already installed.

Just as the IPLV metric (Integrated Part Load Value, AHRI Standards 550/590-2015) [6] was a revolution for air conditioning, the same idea applies to EUED being an evolution for the comparison of where to install a DC. EUED is already a suggestion from ISO 50006 [23] to analyze the energy efficiency of commercial buildings, as it is defined as specific energy (kWh/m$^2$) and not specific power (kW/m$^2$). The initial EUED data can be classified as [22]:

- Free Cooling: a system that uses the enthalpy characteristics of the outside air to acclimate rooms;
- Evaporative: direct or indirect adiabatic cooling using the wet bulb temperature;
- System for geothermal condensation (ground source) as a thermal bath option to condense the refrigerant fluid;
- COP: Coefficient of performance, which is used to evaluate the relationship between the cooling powers obtained and the work spent to obtain it.

The EUED uses, as one of its indicators, the possible use of free cooling and the respective energy consumption.

For each climate and for an entire year, the number of hours characterized by environmental air conditions that allow free cooling is determined. After that, energy consumption is calculated considering these hours and the COP of the multiple solutions such as free cooling, evaporative cooling and refrigeration with varied COP, and it provides a more complete and global view of the whole, as it emphasizes the complete climatic conditions. The EUED given by Equation (4) is used in the design phase of a Datacenter.

$$\text{EUED} = \frac{\text{Total DC Energy with Enthalpy variations [kWh/yr]}}{\text{Specific consumed Energy IT equipment [kWh/yr]}} \tag{4}$$

The EUED methodology uses an average air inlet temperature of 20 °C to specify the following conditions, considering an altitude of 182 m at Chicago (USA) and 908 m at Curitiba (Brazil) [22]:

A. Free Cooling is used when the external air temperature is below 20 °C and the enthalpy is below 42.797 kJ/kg;
B. Evaporative system is used when the temperature is between 15 °C to 24 °C and the enthalpy from 42.7979 kJ/kg to 55.8233 kJ/kg;
C. When the temperature is above 20 °C and enthalpy is above 55.8233 kJ/kg, the normal system is being used under the following conditions:
  1. COP1: Air intake temperature between 24.0 °C and 27.0 °C;
  2. COP2: Air intake temperature between 27.1 °C and 30.0 °C;
  3. COP3: Air intake temperature between 30.1 °C and 33.0 °C;
  4. COP4: Air intake temperature above 33.1 °C;
  5. GEO: if the geothermal temperature is available, it will be used to determine the COP, with a 4 °C differential in the geothermal temperature.

To compare the conditions of COP1, COP2, COP3, and COP4, the CoolPack software [20] was used (started by COP4 to compare a standard isentropic coefficient to be able to identify the COP with the variation in condensation temperatures). For the effect of condensation temperature, the average air intake temperature for each situation added to 11 °C was used as standard, with the conditions shown in Table 4. The results achieved with the software are described below, for the COP4 to COP1 cases in decreasing order, for geothermal water condensation systems the value of 6 °C will be added (the approach value between the air intake temperature in the condenser and the condensation was indicated by the manufacturer Liebert and by Bitzer software) [24].

In all situations, such as those of Santos et al. [25,26], the calculation of COP considered two operative conditions: stopped fans and working fans. Table 5 shows the COP for both operative conditions for each case study.

**Table 4.** Distribution of electricity consumed in a DC.

| Cases | Base Equipment Power (kW) | Condensing Temperature (°C) | Evaporation Temperature (°C) | Cooling Fluid | COP (kW/kW) |
|---|---|---|---|---|---|
| COP1 | 104.5 | 36.5 | 5 | R410 A | 4.381 |
| COP2 | 104.5 | 39.0 | 5 | R 410 A | 4.101 |
| COP3 | 104.5 | 42.5 | 5 | R 410 A | 3.745 |
| COP4 | 104.5 | 44.0 | 5 | R 410 A | 3.633 |

**Table 5.** COP values for different operative conditions (COP just compressors; COP evaporator and condenser with fans.

| Cases | $COP_{Stopped\ fans} = \frac{Refrigeration\ power}{compressors\ work}$ [kW/kW] | $COP_{Working\ fans} = \frac{Refrigeration\ power}{(compressors+fans)\ work}$ [kW/kW] |
|---|---|---|
| COP4 | $\frac{104.5}{21.64} = 4.829$ | $\frac{104.5}{21.64+7.36} = 3.633$ |
| COP3 | $\frac{104.5}{20.54} = 5.087$ | $\frac{104.5}{20.54+7.36} = 3.745$ |
| COP2 | $\frac{104.5}{18.12} = 5.768$ | $\frac{104.5}{18.12+7.36} = 4.101$ |
| COP1 | $\frac{104.5}{16.49} = 6.338$ | $\frac{104.5}{16.49+7.36} = 4.381$ |

These results were used alongside data of the enthalpy system methodology for DCs to build a system of wide psychometric coverage for all possible external temperature points and was elaborated for Free Cooling, Evaporative Cooling or Cooling (COP1 to COP4). The ASHRAE Weather Data Viewer [17] was used to relate dry bulb temperature frequencies with the dew point coincident temperatures. The enthalpy associated with this relationship was found at each point in the following cumulative frequencies for each system (Table 6).

**Table 6.** Data for psychrometric chart for a total thermal load of 790.6 kW.

| System | COP (kW/kW) | Power (kW) |
|---|---|---|
| Free Cooling | 19.180 | 41,220 |
| Evaporative | 16.780 | 47,116 |
| COP1 | 4.381 | 180,462 |
| COP2 | 4.101 | 192,782 |
| COP3 | 3.795 | 211,108 |
| COP4 | 3.633 | 217,616 |

Using the same ASHRAE software [17] and considering the EUED methodology mentioned above, the frequencies of each HVAC solution (free cooling, evaporative cooling and various COPs in refrigeration) were determined for each city (Curitiba and Chicago).

It is worth mentioning that there are differences in the amount received by these cities. In Curitiba, the evaporative cooling system (adiabatic) (temperature between 15°C and 24°C and enthalpy value from 42.7979 kJ/kg to 55.8233 kJ/kg) had a higher number of hours, 3,453,887 h per year, than Chicago, 1,177,678 h a year. With a Free Cooling system (temperature below 20°C and enthalpy below 42.797 kJ/kg), Curitiba has 4410 h per year and Chicago has 6,581,153 h per year.

According to Table 7, the cities of Curitiba and Chicago have different energy consumptions according to their temperature and enthalpy. In a period of 8760 h, Chicago used 0.08% more energy than Curitiba. Table 7 also shows the relationship between COP and energy. The higher the COP, the lower the energy consumption.

Energy consumption with infrastructure, which is the sum of energy consumption with air conditioning, equipment, lighting, and other equipment, between cities, shows a small difference. The indicators obtained with the application of the EUED indicator were, respectively, 1.245 kW/kW for Curitiba and 1.246 kW/kW for Chicago, showing a

difference of 16.86% for Curitiba and Chicago for 18.78% in relation to the PUE COA, as shown in Table 8.

It is important to highlight that the PUE COA and EUED indicators are related to the Data Center as a whole (involving IT equipment, lighting and others). To better visualize the differences exclusively for server cooling, COP PUE COA and COP EUED indicators were created, that is, the ratio of how much thermal energy is taken from a DC by how much specific electrical energy is required for it.

According to Table 9, it is observed that in Chicago the COP EUED is 3.85 times greater than the COP PUE COA, while in Curitiba this difference is 3.61 times.

**Table 7.** Results using Energy Usage Effectiveness Design (EUED) indicator rules (adapted from [21]).

| System | Total Thermal Load (kW) | COP (kW/kW) | Power (kW) | $h_{Curitiba}$ (Hours) | $h_{Chicago}$ (Hours) |
|---|---|---|---|---|---|
| Free Cooling | 709.6 | 19.180 | 36.997 | 4410 | 6581 |
| Evaporative | 709.6 | 16.781 | 42.288 | 3454 | 1178 |
| COP1 | 709.6 | 4.381 | 161.972 | 580 | 467 |
| COP2 | 709.6 | 4.101 | 173.031 | 270 | 379 |
| COP3 | 709.6 | 3.795 | 189.479 | 46 | 122 |
| COP4 | 709.6 | 3.633 | 195.321 | 0.5 | 33 |
| System | Energy Chicago Air (kWh/year) | Energy Curitiba Air (kWh/year) | Equipment IT (kWh/year) | Lighting (kWh/yea) | Others (kWh/year) |
| Free Cooling | 243,482.734 | 163,156.204 | 6,084,696 | 140,160 | 420,480 |
| Evaporative | 49,802.164 | 146,059.486 | | | |
| COP1 | 75,700.115 | 93,902.222 | | | |
| COP2 | 65,559.011 | 46,696.559 | | | |
| COP3 | 23,057.168 | 8713.585 | | | |
| COP4 | 6490.506 | 99.418 | | | |

**Table 8.** Final results using EUED indicator rules.

| City | Energy (kWh/yr) | PUE COA (kW/kW) | EUED (kWh/yr)/(kWh/yr) | Difference between PUE COA and EUED (%) |
|---|---|---|---|---|
| Curitiba (Brazil) | 7,109,427.698 | 1.497 | 1.245 | 16.86% |
| Chicago (USA) | 7,103,963.236 | 1.498 | 1.246 | 18.78% |

**Table 9.** Comparison of PUE COA, EUED, COP PUE COA and COP EUED.

| City | PUE COA | EUED | COP PUE COA | COP EUED |
|---|---|---|---|---|
| Curitiba (Brazil) | 1.455 | 1.245 | 3.745 | 13.553 |
| Chicago (USA) | 1.480 | 1.246 | 3.477 | 13.394 |

The expression of COP EUED is:

$$\text{COP EUED} = \frac{\text{Energy input HVAC year [kWh]}}{\text{Average thermal load} \cdot 8760 \text{ h}} \tag{5}$$

As can be seen, the cities of Curitiba and Chicago have a similar COP EUED. In general, Chicago is known as a windy city, characterized by snow and cold. It has characteristics, even using free cooling, that are very close to the characteristics of a city as populous as Curitiba in south Brazil. Thus, in terms of energy comparisons with an indicator such as the EUED, the difference is small, that is, inconsiderable.

*Comparison with TEWI DC Metric*

The application of TEWI with the COP EUED is given by Equation (6) and provides a version of the TEWI indicator for DC, TEWI DC, for a 10-year life cycle [27]:

$$\text{TEWI DC} = \text{GWP(direct leaks including EOL)} + \left( \frac{Q \cdot 8760 \text{ hr}}{\text{COP EUED}} \right) \cdot \beta \cdot 10 \tag{6}$$

where:

Q = Average thermal load (kW).

COP EUED = Coefficient of Performance Energy Usage Effectiveness Design (kW/kW).

Specifically, for direct emissions, the fluid R 410 A (indicated by the manufacturer) with GWP of 2088 was used, with a refrigerant weight of 33.17 kg per equipment (as there are 8 machines, the total value was 265.38 kg). In a period of 10-year life, a leak rate of 12.5% per year ($L_{annual}$), and a recovery of 70% ($\alpha_{recovery}$) of the refrigerant fluid was considered, for direct emissions. The same values are used for both cities. Applying Equation (2), the component of direct emissions was 848,823.63 (kg $CO_2$) [27,28].

According to Table 10, despite the similar annual power consumption of DC located in these cities, their TEWI values are extremely different. TEWI for Chicago (USA) is 2,784,102.130 (kg $CO_2$/10 yrs) and for Curitiba (Brazil) it is 1,252,409.640 (kg $CO_2$/10 yrs). The installation and operation of a DC in Chicago lead to much higher $CO_2$ emissions (more than x2.22) than in Curitiba. The values of TEWI DC (indirect and direct $CO_2$ emissions) are shown in Table 11 and compared in Figure 2 using updated 2019 data from Brazil and the USA. This difference is justified by the clean energy matrix that Brazil has, which is basically hydroelectric. Thus, from the indirect emission component of Brazil arises an emission of 0.088 (kg $CO_2$/kWh), while in the USA it is determined tp be an emission of 0.417 (kg $CO_2$/kWh).

**Table 10.** TEWI comparison between cities for DC installation.

| City | EUED | COP PUE COA | Consumed Energy HVAC | TEWI DC |
|---|---|---|---|---|
| Curitiba (Brazil) | 1.245 | 3.745 | 510,979.200 | 1,252,409.640 |
| Chicago (USA) | 1.246 | 3.477 | 517,066.640 | 2,784,102.130 |

**Table 11.** TEWI DC (Indirect and direct $CO_2$ emissions) comparison.

| City | Indirect $CO_2$ Emissions | Direct $CO_2$ Emissions | TEWI DC |
|---|---|---|---|
| Curitiba (Brazil) | 403,586 | 848,823.600 | 1,252,410 |
| Chicago (USA) | 1,935,279 | 848,823.600 | 2,784,102 |

The impact of refrigerant fluid in Curitiba represents 68% of emissions and the energy matrix only 32%, while in Chicago the impact of the refrigerant fluid represents 30%, while the energy matrix represents 70%, this result means that different air conditioning systems will have different responses in each city. That is, in Brazil it weighs more in decision making with clean matrix systems that have less refrigerant charge and a lower GWP. In Chicago, energy efficiency is more important and a high efficiency system with VRF may be interesting in polluting energy matrices, while a Chiller system with less technology may be interesting in a clean energy matrix location.

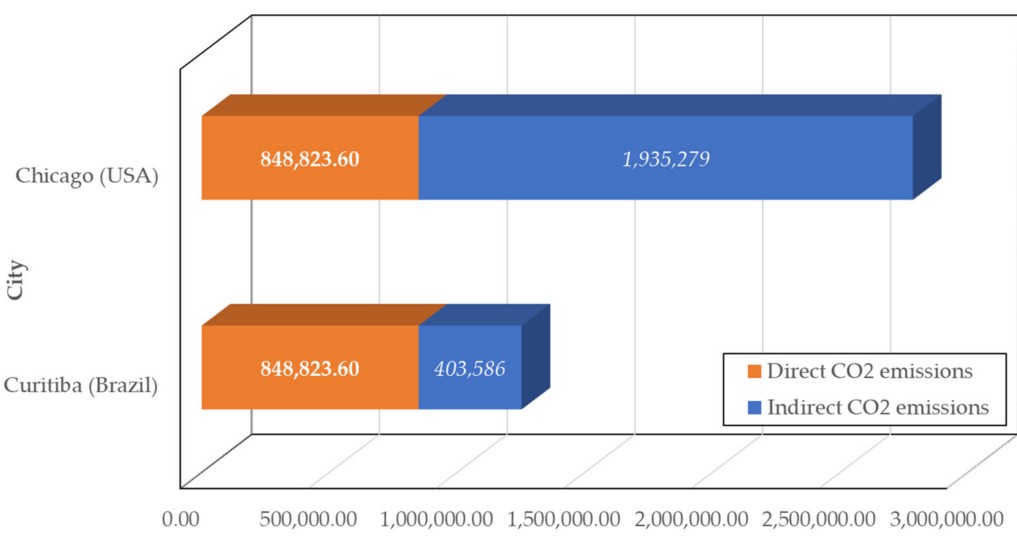

**Figure 2.** Comparison of TEWI DC (direct and indirect $CO_2$ emissions) of a DC installed in Chicago and Curitiba.

## 5. Conclusions

TEWI is a scientific method already consolidated to measure the environmental impact in HVAC&R systems [29]. With the support of the EUED concept, a new methodology named TEWI DC was used to measure the environmental impact in these systems, considering 8760 h, including the lifetime use of equipment. A calculation was performed comparing two cities with similar thermal characteristics but with different energy matrices. It was concluded that a DC in Chicago would generate 2.22 times more of an environmental impact than Curitiba (the same kind of air conditioning equipment).

The question of the holistic view of TEWI DC is an important indicator in decision making, not only based on where to install a DC (the question most addressed by the EUED), but also in terms of what type of refrigerant to choose and the energy input matrix that will be used in a building, and this methodology can be replicated in multiple cities and with different air conditioning systems.

With such importance placed on the environment and $CO_2$ emissions nowadays, it is essential to rely on LEED or AQUA certifications, and indicators such as TEWI DC for prior qualification for certification. In the scores, this indicator is sensitive and helps gain a more systemic view.

As Lord Kelvin said [30]: "If you cannot measure it, you cannot improve it . . . When you can measure what you are speaking about, and express it in numbers, you know something about it; but when you cannot measure it, when you cannot express it in numbers, your knowledge is of a meager and unsatisfactory kind".

**Author Contributions:** Conceptualization, A.F.S. and P.D.G.; methodology, A.F.S.; validation, A.F.S. and H.J.L.d.S.; formal analysis, A.F.S. and P.D.G.; investigation, A.F.S.; resources, H.J.L.d.S.; data curation, A.F.S. and P.D.G.; writing—original draft preparation, A.F.S., H.J.L.d.S.; writing—review and editing, P.D.G.; visualization, H.J.L.d.S.; supervision, P.D.G.; project administration, A.F.S. All authors have read and agreed to the published version of the manuscript.

**Funding:** This research received no external funding.

**Institutional Review Board Statement:** Not applicable.

**Informed Consent Statement:** Not applicable.

**Data Availability Statement:** Data can be found in the references cited in the manuscript.

**Conflicts of Interest:** The authors declare no conflict of interest.

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
