# Peer review of "Ecoenergetic Comparison of HVAC Systems in Data Centers"

_climate, doi:10.3390/cli9030042_

Round 1

Reviewer 1 Report

The authors have improved their manuscript and addressed my all comments in revised version. I do not have anything against publishing this paper.

Author Response

have 

Reviewer 2 Report

The subject of the paper is interesting, because the theme of environmental impact due to refrigerants and cooling systems is a current issue. Moreover, to provide simple indexes is a very evaluable effort, whenever such indexes simplify the work of decision makers.

Unfortunately, the paper needs a strong revision beacuse: entire periods are repeated, some sentences are abruptly interrupted, concepts and methods are not clearly explained, and there is a lack of consequentiality.

Language must be also strongly improved, as in many parts, due to syntactic errors, it is very difficult to understand phrases meaning.

In the following lines, some of the main issues are highlighted.

Abstract

The considered issue and the lack of knowledge must be better explained.

Please, explain a little better the index proposal.

1 Introduction

The term simulation is used instead of calculation, relating to GPW index, why?

From 86 to 97 a revision of the speech must be carried out, there are more than one repetitions of the same concept.

Line 97, second part, seems not to belong to the main phrase.

2 Methodology

The paragraph need a deep revision. The speech must be organized in a more logical and comprehensible form.

2.1 Data

This part must be improved. It is very difficult to understand the meaning

from line 103 to line 109 seems a draft speech to be better developed. The speech is reprised in the paragraph 2.12.
From line 110 it is introduced soft drink selection process, but it is not explained why.

2.1.1 TEWI

Line 143: it is not clear of what results the sentence is talking about.

Lines 146 and 147 are not clear

2.12

Please, better explain why TEWI is used instead of GWP.

2.2.1

Line 166, please, explain the term “climate simulation”.

How the CO2 emission calculation was carried out? It seems that it takes into account only the GWP and not the consumed electricity. Why? Please explain better the Table 1. It seems only a GWP estimation and a translation into equivalent CO2 emission.

It was said (line 152, 153) “The most significant index for getting reliable sources for low-impact air-conditioning and cooling solutions is the GWP. Though, TEWI metric is used in this study to identify the demand in the Brazilian market.” Thus it is not clear what has actually been done.

Maybe it was more correct to say “In the current paper, not only GWP, but also TEWI was calculated”

3 Case study

Table 2 would be more explanative if jointed with a graph that shows the average monthly behaviour of: dry bulb temperature, relative humidity and enthalpy.

Line 222: sensible, not sensitive heat!

Form line 221 to line 225 Heat loads must be represented in a more intelligible way, e.g. by means of a table. It is not specified how many people are involved and the related heat load (sensible and latent).

Table 3: sensible cooling, not sensitive!

4 Comparison

The paragraph need a deep revision. The speech must be organized in a more logical and comprehensible form.

Lines 252 and 253 are very difficult to be understood.

“The value of the PUE COA, that is, the energy efficiency metric regarding the use of

Constant Outdoor Air (COA) temperature at an average of 0.4% of the current highest temperatures of NBR 16401, 2017 [14]”.

Please, provide the EUED formula and a short explanation, as done for TEWI

Lines 258 and 259 are very difficult to be understood.

From line 265 to line 274, the speech is not clear. Please, provide a better explanation of the indexes (better to say indicators). For example, it can be said something similar to: “the EUAD uses, as one of its indicators, the possible use of free cooling and the related energy consumption.

For each climate and for an entire year the number of hours characterised by environmental air conditions that permit free cooling is determined. After that, energy consumption is calculated, taking into account those hours and the COP of the free cooling solution…. Etc.”

Form line 292 to line 294 it is declared:

“For the effect of condensation temperature, the average air intake temperature for each situation added to 11°C was used as standard, with the conditions shown in Table 4.”

Please, justify the choice.

It is very difficult to understand the speech from 304 to 310

Table 8 shows differences between PUE COA and EUED and not between PUE and EUED as declared at lines 327 and 328. Moreover, at line 328 the percentages, coming from the declared comparison between PUE and EUED are wrong.

It would be advisable to declare that the percentage is calculated considering, at the denominator, EUAD.

From line 331 to line 335: please control the speech, it is incomprehensible.

How the COP EUED was calculated?

4.1

From line 345 to line 347, please, be clearer: what is, exactly, the applied methodology? Please, explain it step by step.

In table 10 appears “COP PUE COA /Fan”, please, explain such a voice-

Please, provide here, not in conclusions, the discussion for TEWI results.

Please, repeat beta factor values

Conclusions

The reference must be transported in the paragraph where TEWI is presented.

The speech from line 376 to line 379 must be written in a clearer form.

Author Response

ok

Reviewer 3 Report

The authors are presenting a methodology TEWI DC to access the climate impact of HVAC system in Data Centers.

The paper is of interest for the journal scope. There are major changes that should be addressed prior the manuscript acceptance.

  • Review the manuscript template
  • Add a Nomenclature
  • Change the title. The title does not seem appropriate.
  • Abstract should be rewritten. Clarify better the innovation of this paper in the abstract and in the main text.
  • Explain the second term (GWP x m x (1-alpha) of equation 2
  • Explain the choice of Chicago and Curitiba.
  • What are the benefits of the results in a global context? Please explain this better in the manuscript.

Author Response

ok

Round 2

Reviewer 2 Report

The paper was improved and now it is more intelligible.
From my point of view, in order to be clearer it must be added a short scheme, in introduction, briefly describing the various steps of the work
There are other few issues:
From line 187 to line 189, it would be advisable to give a hint here about how much TEWI would be a better tool, more complete, in comparison to GWP. 
In tab 8 PUE COA values are missing. Please add them before the difference percentage column.
There are two tables 9
Table 11 has the same error of the previous version (two times repeated direct CO2 emissions)
There are two conclusion chapters. Please, provide only one of them, rearranging the speech in a more logic manner. 

Author Response

Response to Editor:

We would like to thank the Editor and Reviewers for carefully examining our work and for providing us with the opportunity of revising and improving the manuscript. We have addressed all the comments and suggestions of Reviewers and modified the paper accordingly. All modifications are marked in blue colour in the revised manuscript in order to facilitate the review process. Thank you very much. The improvements made in the paper are listed below.

Please see below our detailed responses to every single comment raised. The authors are thanking again for the attention given to this work, we look forward to hearing from you soon.

Yours sincerely,

The Authors

Reviewer 2/Comment 1: The paper was improved and now it is more intelligible. From my point of view, in order to be clearer it must be added a short scheme, in introduction, briefly describing the various steps of the work

Answer: The authors of the paper thank you for this comment. Your support significantly strengthened the paper content.

Reviewer 2/Comment 2: From line 187 to line 189, it would be advisable to give a hint here about how much TEWI would be a better tool, more complete, in comparison to GWP.

Answer: The authors of the paper thank you for this comment, this emphasis is very important, but it has already been included in lines 134-136.

Reviewer 2/Comment 3: In tab 8 PUE COA values are missing. Please add them before the difference percentage column.

Answer: The authors of the paper thank you for this comment, PUE COA was added in the column

Reviewer 2/Comment 4: There are two tables 9

Answer: The authors of the paper thank you for this comment. The tables have been renumbered.

Reviewer 2/Comment 5: Table 11 has the same error of the previous version (two times repeated direct CO2 emissions)

Answer: The authors of the paper thank you for this comment. The error was corrected.

Reviewer 2/Comment 6: There are two conclusion chapters. Please, provide only one of them, rearranging the speech in a more logic manner.

Answer: The authors of the paper thank you for this comment, a rearrangement was made at completion as requested, seeking logical order.

Reviewer 3/Comment 1: Thank you for considering my suggestions.

Answer: The authors of the paper thank you for this comment. Your support significantly strengthened the paper content.

Reviewer 3 Report

Thank you for considering my suggestions.

Author Response

Response to Editor:

We would like to thank the Editor and Reviewers for carefully examining our work and for providing us with the opportunity of revising and improving the manuscript. We have addressed all the comments and suggestions of Reviewers and modified the paper accordingly. All modifications are marked in blue colour in the revised manuscript in order to facilitate the review process. Thank you very much. The improvements made in the paper are listed below.

Please see below our detailed responses to every single comment raised. The authors are thanking again for the attention given to this work, we look forward to hearing from you soon.

Yours sincerely,

The Authors

Reviewer 2/Comment 1: The paper was improved and now it is more intelligible. From my point of view, in order to be clearer it must be added a short scheme, in introduction, briefly describing the various steps of the work

Answer: The authors of the paper thank you for this comment. Your support significantly strengthened the paper content.

Reviewer 2/Comment 2: From line 187 to line 189, it would be advisable to give a hint here about how much TEWI would be a better tool, more complete, in comparison to GWP.

Answer: The authors of the paper thank you for this comment, this emphasis is very important, but it has already been included in lines 134-136.

Reviewer 2/Comment 3: In tab 8 PUE COA values are missing. Please add them before the difference percentage column.

Answer: The authors of the paper thank you for this comment, PUE COA was added in the column

Reviewer 2/Comment 4: There are two tables 9

Answer: The authors of the paper thank you for this comment. The tables have been renumbered.

Reviewer 2/Comment 5: Table 11 has the same error of the previous version (two times repeated direct CO2 emissions)

Answer: The authors of the paper thank you for this comment. The error was corrected.

Reviewer 2/Comment 6: There are two conclusion chapters. Please, provide only one of them, rearranging the speech in a more logic manner.

Answer: The authors of the paper thank you for this comment, a rearrangement was made at completion as requested, seeking logical order.

Reviewer 3/Comment 1: Thank you for considering my suggestions.

Answer: The authors of the paper thank you for this comment. Your support significantly strengthened the paper content.

This manuscript is a resubmission of an earlier submission. The following is a list of the peer review reports and author responses from that submission.

Round 1

Reviewer 1 Report

The paper deals with interesting and new issue- ecoenergetic simulation of HVAC systems in Data Centers based on Brasilian example. However paper lack literature review and background. The list of references is pour. It needs to be updated by extending critical literature review on the subject. Also discussion section is missing. It  should be added and results need to be discussed with other studies findings. Conclusions are weak. Limitations of the study need to be addressed. The future research guidelines and policy implications as well. The input of paper needs to be stressed.  Paper needs major revision to be published. 

Reviewer 2 Report

  1. The paper contains too many abbreviations in the abstract, such as HCFC, TEWI etc, which need to be explained from the beginning.
  2. Main research method is missing the abstract and the section 2.
    The literature review of existing work and studies are missing in the introduction.
  3. As a result, the research motivation and novelty are not clear.
  4. The reasons to perform cast studies in only two cities are not presented.
  5. TEWI is not the only index for data centers, why the authors only consider TEWI?
  6. Validation of the simulation result is missing.
  7. Sensitivity analysis is missing.
  8. Discussion about the simulation results are not fully presented.
  9. The connection with climate journal is weak.
  10. The overall paper is more like a technical report, rather than a scientific paper.